# Random Spiking Neural Networks are Stable and Spectrally Simple

Ernesto Araya[1], Massimiliano Datres[*1], and Gitta Kutyniok[1,2,3,4]

[1]Ludwig-Maximilians-Universität München
[2]Munich Center for Machine Learning (MCML)
[3]University of Tromsø
[4]DLR-German Aerospace Center
{araya, datres, kutyniok}@math.lmu.de

## Abstract

Spiking Neural Networks (SNNs) offer energy-efficient computation. We analyze discrete-time Leaky Integrate-and-Fire (LIF) SNNs via Boolean function theory, characterizing their *noise sensitivity* and *stability* under input perturbations. We show that wide LIF-SNN classifiers are on average stable, with Fourier spectra concentrated on low frequencies, implying a bias toward simple functions. Experiments confirm that LIF SNNs exhibit stability.

## 1 Introduction

Artificial Neural Networks (ANNs) power modern ML but incur high computational and energy costs [1]. Spiking Neural Networks (SNNs), operating through event-driven spikes, offer a biologically inspired, energy-efficient alternative well suited for neuromorphic hardware [2, 3]. Yet theoretical understanding—especially of stability and generalization—remains limited. We analyze discrete-time Leaky Integrate-and-Fire (LIF) SNNs using Boolean function theory [4], providing the first characterization of SNNs stability via noise sensitivity. This links stability to *spectral simplicity*, the concentration of the Fourier–Walsh spectrum on low frequencies, reflecting a bias toward simple functions [5, 6]. Random LIF networks [7] naturally exhibit this property.

**Contributions.** (1) Probabilistic bounds show random LIF-SNNs are stable to $\mathcal{O}(\sqrt{n})$ input perturbations [8]. (2) We formalize and prove their bias toward spectrally simple functions. (3) Empirical results confirm that SNNs exhibit noise stability.

**Related Works** SNN stability has been analyzed via Lyapunov methods in ANNs and Neural ODEs [9–11] and through robustness bounds for LIF models [12]. We instead study classifier-level stability under reset-by-subtraction using Boolean function analysis [4, 8]. Simplicity bias has been linked to

generalization [5, 6, 13, 14]. We propose *spectral simplicity*, a Fourier–Walsh notion naturally fitting SNNs.

**Notation** For $a \in \mathbb{N}$, let $[a] = \{1, \ldots, a\}$. For $x, y \in \{-1, 1\}^n$, $d_H(x, y) = \frac{1}{2}\sum_i |x_i - y_i|$, and $\operatorname{sign}(x) = 1$ if $x \geq 0$, $-1$ otherwise.

We write $\mathcal{N}(u, \Sigma)$ for Gaussian, $\operatorname{Unif}(\{-1, 1\}^n)$ for uniform on the hypercube, $\operatorname{Bin}(n, p)$ for Binomial, $\operatorname{Rad}(\kappa)$ for i.i.d. Rademacher$(\kappa)$, and use $\mathcal{O}(\cdot), o(\cdot), \omega(\cdot)$ asymptotics.

## 2 The Discrete-Time LIF Model

We study the stability of *sign leaky integrate-and-fire* (sLIF) spiking neurons, which map time-series inputs $(x_t)_{t \in [T]} \in (\{-1, 1\}^n)^T$ to binary spikes $(s_t)_{t \in [T]} \in \{-1, 1\}^T$ via

$$\begin{cases} u_t = \beta u_{t-1} + w^\top x_t - \frac{\theta}{2}(s_{t-1} + 1), \\ s_t = \operatorname{sign}(u_t - \theta), \quad u_0 = 0, \end{cases} \tag{1}$$

where $u_t \geq 0$ is the membrane potential, $w \in \mathbb{R}^n$ the weights, $\beta \in [0, 1]$ the leak, and $\theta > 0$ the threshold. When $u_t > \theta$, the neuron spikes ($s_t = 1$) and resets by subtraction. Weights are initialized as $w \sim \mathcal{N}(0, I_n/n)$ to keep activations $\mathcal{O}(1)$. LIF neurons generalize integrate-and-fire units ($\beta = 1$) [15].

**Networks.** An $L$-layer sLIF SNN with widths $n_0, \ldots, n_L$ and latency $T$ evolves as

$$\begin{cases} u_t^{(l)} = \beta u_{t-1}^{(l)} + W^{(l)} s_t^{(l-1)} - \frac{\theta}{2}(s_{t-1}^{(l)} + 1), \\ s_t^{(l)} = \operatorname{sign}(u_t^{(l)} - \theta), \ s_t^{(0)} = x_t, \ u_0^{(l)} = 0, \end{cases} \tag{2}$$

with $W^{(l)} \in \mathbb{R}^{n_l \times n_{l-1}}$. The output class is predicted via spike counts:

$$f^{L,T}\big((x_t)_{t \in [T]}, W\big) = \arg\max_{i \in [n_L]} \sum_{t=1}^{T} s_{t,i}^{(L)}.$$

**Assumptions** We assume $n_0 = \cdots = n_{L-1} = n$, $n_L$ equal to the number of classes, and $\beta = 1$. Parameters are initialized as $W_i \sim \mathcal{N}(0, 1/n)$.

---

*Corresponding Author.

## 3 Noise Sensitivity of Boolean Functions

sLIF neurons can be viewed as compositions of Boolean functions, enabling analysis via Boolean function theory [4]. For $f : \{-1, 1\}^n \to \{-1, 1\}$ and noise rate $\nu$, the *noise sensitivity*

$$\mathbf{NS}_\nu(f) = \mathbb{P}_{x,\xi}[f(x) \neq f(x \odot \xi)],$$

with $x \sim \text{Unif}(\{-1, 1\}^n)$ and $\xi_i \sim \text{Rad}(1 - \nu)$, quantifies the probability that $f$ flips under random bit perturbations. Related to noise sensitivity, *noise stability*, defined as $\mathbf{Stab}_{1-2\nu}(f) = 1 - 2\,\mathbf{NS}_\nu(f)$, measures the correlation between $f(x)$ and $f(x \odot \xi)$. For a family $\{f_w\}$ with $w \sim \mu$,

$$\mathbf{ENS}_\nu = \mathbb{P}_{w,x,\xi}[f_w(x) \neq f_w(x \odot \xi)].$$

Each $f$ admits a Fourier–Walsh expansion $f(x) = \sum_{S \subseteq [n]} \hat{f}(S)\chi_S(x)$, and is said to be $\epsilon$-*concentrated up to degree* $k$ if $\sum_{|S|>k} \hat{f}(S)^2 \leq \epsilon$. In expectation, $\mathbb{E}_{w \sim \mu}\sum_{|S|>k}\hat{f}_w(S)^2 \leq \epsilon$. By [4, Prop. 3.3], $\epsilon = 3\mathbf{NS}_\nu(f)$ implies concentration up to degree $1/\nu$. Thus, small noise sensitivity corresponds to spectral simplicity, a property we later show for wide sLIF networks.

## 4 Noise stability of SNN classifiers

We analyze the stability of LIF-SNN classifiers using Boolean function tools from Section 3. Each output $s_t$ of a single sLIF neuron defines a Boolean function $s_t : \{-1, 1\}^n \to \{-1, 1\}$.

**Theorem 1** (Single neuron stability). *Let* $w \sim \mathcal{N}(0, I_n/n)$ *and* $x_t, y_t \in \{-1, 1\}^n$ *with* $\nu_t = d_H(x_t, y_t)/n$, $\bar{\nu}_t = \frac{1}{t}\sum_{k=1}^{t} \nu_k$. *If* $\max_t \nu_t = \mathcal{O}(1/\sqrt{n})$, *then for all* $t \in [T]$,

$$\mathbb{P}_w[s_t(x_1, \ldots, x_t) \neq s_t(y_1, \ldots, y_t)]$$
$$\leq C(1 + \theta)t^2\sqrt{\bar{\nu}_t}\log n,$$

*with* $C > 0$ *absolute; for static inputs the* $\log n$ *factor can be removed.*

Extending this argument to networks of sLIF neurons, we obtain the following multi-layer stability result.

**Theorem 2** (Multi-layer SNN stability). *For an $L$-layer sLIF network classifier* $f^{L,T}(\cdot, W)$ *with* $n_1 = \cdots = n_{L-1} = n$ *and* $W \sim \mathcal{N}(0, I_d/n)$, *let* $\nu = \max_t \nu_t = \mathcal{O}(1/\sqrt{n})$. *For large* $n$,

$$\mathbb{P}_W[f^{L,T}(x_1, \ldots, x_T) \neq f^{L,T}(y_1, \ldots, y_T)]$$
$$\leq n_L T^4 C(1+\theta)\nu^{1/2^{2L+1}}\log^{3/2} n$$
$$+ (L-1)e^{-c\nu^{1/2^{2L-1}}n},$$

*with absolute constants* $C, c > 0$.

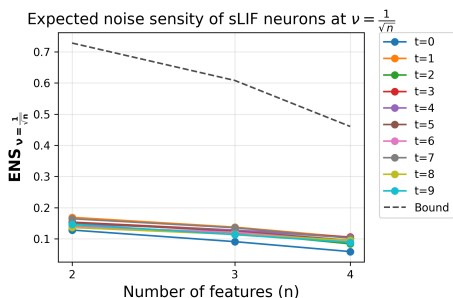

**Figure 1.** Noise sensitivity $\mathbf{ENS}_{1/\sqrt{n}}$ for different input dimensions $n$ for sLIF with $\theta = 0.5, \beta = 1.0$ and $T = 10$.

Building on Theorem 2, for a binary classifier $f^{L,T}(\cdot, W)$ ($n_L = 2$) with static inputs (constant input over time) the following result holds.

**Corollary 3** (Expected noise sensitivity). *Let* $\nu' \leq 1/\sqrt{n \log n}$, *we have*

$$\mathbf{ENS}_{\nu'}(\{f^{L,T}\}) \leq C_{T,\theta}\nu'^{1/2^{2L+1}}\log^{3/2} n$$
$$+ (L-1)e^{-c\nu'^{1/2^{2L-1}}n} + e^{-\frac{1}{4}\sqrt{n}},$$

*and the expected Fourier spectrum is $\epsilon$-concentrated up to degree $1/\nu'$ with* $\epsilon = C_{T,\theta}\nu'^{1/2^{2L+1}}\log^{3/2} n$.

Choosing $\nu' = 1/\sqrt{n \log n}$ shows that only a vanishing fraction of degrees contribute meaningfully, so LIF-SNN classifiers are spectrally simple. Concentration deteriorates with $L, T$, and $\theta$; the $L$-dependence is intrinsic to Boolean compositions, while the $\theta$- and $T$-dependencies likely stem from proof artifacts.

## 5 Experiments

We empirically evaluate the upper bound on $\mathbf{ENS}_\nu$ presented in Theorem 1 under static inputs. For single sIF and IF neurons ($n = 100$–$10^4$, $\theta = 0.5$, $T = 10$), Monte Carlo estimates (averaged over 10 weight samples and 100 perturbations) show consistently low sensitivity across $t$.

## 6 Conclusion

We studied the stability of wide SNN classifiers via Boolean analysis, giving bounds on expected noise sensitivity and linking to spectral simplicity. Most assumptions can be relaxed; large widths are required for concentration, while input distribution can be generalized without affecting Corollary 3.

**Future directions.** Extend to general SNNs, explore alternative perturbations (adversarial robustness), analyze average distance to label flips [5], and study initialization effects on stability.

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
