# OpenReview forum: "Random Spiking Neural Networks are Stable and Spectrally Simple"
_NLDL.org/2026/Abstracts_Track — NLDL 2026 Abstracts_

### Official Review · Reviewer_g6rA · 2025-10-27

**Soundness:** 2
**Correctness:** 3
**Rating:** 4
**Confidence:** 3

**Summary:**

This paper focuses on an analysis of discrete-time Leaky Integrate-and-Fire (LIF) Spiking neural networks using Boolean function theory to analyse their sensitivity to noise as well as their stability under input perturbations.
LIF SNN map time-series inputs to binary spikes, and the equation for this can be generalized to L dimensional layers. Boolean analysis is able to be done as sLIF neurons can be viewed as a composition of Boolean functions.
This work is entirely theoretical and through their analysis, they see that they can obtain probabilistic bounds which shows how the data is stable to the bounds.

**Strengths:**

-	Notation seems to be clear
-	The paper presentation seems to be clear with the different sections.
-	Theoretical analysis of spiking neural networks are still an area which could be important to look into.

**Weaknesses:**

-	There does not seem to be much motivation for the work. There only seems to be a few sentences to discuss the benefits of spiking neural networks, which makes it of interest.
-	Additionally, there does not seem to be much discussion on why it would be beneficial if the spiking neural network is stable.

---

### Official Review · Reviewer_okTb · 2025-10-28

**Soundness:** 3
**Correctness:** 3
**Rating:** 4
**Confidence:** 3

**Summary:**

This paper analyzes the stability properties of Spiking Neural Networks (SNNs), specifically discrete-time Leaky Integrate-and-Fire (LIF) models, using Boolean function theory.

**Strengths:**

- This appears to be the first work to analyze SNN stability through Boolean function theory and noise sensitivity, in contrast to traditional Lyapunov methods.
- The paper provides formal theorems with probabilistic bounds (Theorems 1-2 and Corollary 3) that characterize stability properties.
-  By linking stability to spectral simplicity, the work provides theoretical insight into why SNNs might generalize well.

**Weaknesses:**

It is possible that the actual bounds obtained may not be tight or useful and lack of complete proofs, makes the reading difficult ;
However for an abstract with page limits, this meets expectation.

---

### Decision · Program_Chairs · 2025-11-05

**Decision:**

Accept

**Comment:**

The abstract is of interest to the community and should be presented at the conference.